# Environmental Regulation and Chronic Conditions: Evidence from China’s Air Pollution Prevention and Control Action Plan

**DOI:** 10.3390/ijerph191912584

**Published:** 2022-10-02

**Authors:** Yang Zhao, Beomsoo Kim

**Affiliations:** 1School of Economics and Management, Yanshan University, Qinhuangdao 066004, China; 2Department of Economics, Korea University, Seoul 02841, Korea

**Keywords:** fine particulate air pollution (PM_2.5_), Air Pollution Prevention and Control Action Plan, chronic health conditions, difference-in-differences model

## Abstract

In January 2013, a dense haze covered 1.4 million kilometers of China and affected more than 800 million people. Air pollution in China had become a serious threat to the daily lives of people. The State Council of China enacted the “Air Pollution Prevention and Control Action Plan” (APPCAP) in 2013 to lower the particulate matter (PM) level. Between 2013 and 2017, each administrative division established its own environmental preservation strategy in accordance with the APPCAP. We examined the effects of the nationwide air pollution control policy, APPCAP, on chronic health conditions among adults using a nationally representative survey, CFPS, conducted in 2012, 2014, and 2016. We applied a difference-in-differences model, using the time gap when each administrative division implemented the APPCAP. We found that the APPCAP significantly reduced doctor-diagnosed chronic conditions of the respiratory and circulatory systems in the last six months. In respiratory diseases and circulatory system diseases, the treatment effect of the APPCAP was a 34.6% and 11.5% reduction in the sample mean, respectively. The poorest socioeconomic groups and the elderly benefited the most. The stronger the goal, the more positive the effects were on health; the longer the policy intervention, the better the health outcomes were.

## 1. Introduction

The World Health Organization (WHO) reported that outdoor air pollution was responsible for the premature deaths of some 3.7 million people in 2012, most of them in low- and middle-income countries [1]. Exposure to ambient air pollution increases mortality and morbidity and is a leading cause of the global disease burden [2].

China has experienced rapid economic growth over several decades. At the same time, particulate air pollution caused by the coal-based energy-intensive development path and increase in motor vehicle use has become serious [3]. Previous studies have shown that atmospheric particulate pollution ranks fourth among the twenty major factors causing fatal harm to the public in China [4]. In January 2013, a dense haze covered 1.4 million kilometers of China and affected more than 800 million people [5]. In the national capital region, the annual average concentration of PM_2.5_ is 93 μg/m^3^ [6], which is almost 10 times higher than the WHO standard (10 μg/m^3^) [7]. Rohde and Muller (2015) estimated that 92% of the population of China experienced unhealthy air based on the U.S. Environmental Protection Agency standards during their study and calculated that 1.6 million deaths/year in China were contributed to by air pollution, roughly 17% of all deaths in China [8].

China is at its most important stage of ambient air pollution control, and the unaddressed health consequences at regional and global levels pose major policy challenges [9]. The State Council of China enacted the APPCAP in 2013 to lower the particulate matter (PM) level. Between 2013 and 2017, each administrative division established its own environmental preservation strategy in accordance with the APPCAP. This policy is quite stringent, which may be a milestone [10]. Considering the concern about air pollution in China and the threats to people’s health, the APPCAP provides an excellent opportunity to evaluate the benefits of the air pollution control policy.

Although there is some literature showing that air pollutants contribute to increased mortality and hospital admissions [11,12,13], limited empirical studies evaluate the benefits to health of a reduction in air pollution. Increased air pollution and decreased air pollution may have asymmetrical effects on health. A few papers examining the benefit of air quality improvement in China focused on specific areas in China [14,15] or measured the extreme outcome of mortality changes over time [3]. Mortality is the most studied health endpoint in association with air pollution. One reason is the widespread availability of data for large populations, and another reason is the importance of mortality in estimating health impacts [16]. However, observing mortality over time can be confounded with other factors including medical technology innovations [17]. By focusing on specific areas, we might miss the negative spillover effect where air pollution lowers in the target area but worsens in the neighboring area to which factories move [18]. Some studies have also been conducted using the special air pollution control for specific events that have occurred in China, for example, the effect of the Beijing Olympic air quality intervention on cardiovascular diseases [14], pulmonary inflammation in children [19], and heart rate variability in taxi drivers [15], the APEC Blue and Parade Blue periods in Beijing [20], the 2010 Guangzhou Asian Games [21], and the 2010 Shanghai World Expo [22]. Although evaluating temporary interventions might be useful to measure health benefits; this temporary intervention may not be applicable for long period of time.

Air pollution in China may impact people through both acute (peak level) and chronic (average level) exposures. Severe particulate pollution days (SPPDs), a chronic exposure measure, have been observed frequently in China, since air pollution remains for an extended period of time [23]. As a result, air pollution may impact people’s chronic conditions, first and foremost. In this paper, we aim to evaluate the effects of the APPCAP policy, which is the first nationwide air quality improvement policy in China, on chronic conditions using difference-in-differences estimation method for all of China. Each administrative division implemented the APPCAP policy at a different time, and we will use this implementation time gap to estimate the causal inference of the policy on health.

This paper makes several contributions to the literature. First, we measure air-pollution-related chronic disease as an outcome. Many earlier studies instead focused on death [3,24,25]. Although death is a clear and catastrophic event, chronic diseases that affect people’s daily lives and quality of life are also important to investigate. Second, we study adults as a sample. There are few publications examining adults, since adults with fully functional organ capacity (for example, lung) are less vulnerable compared to children or adolescents [26]. Third, we evaluate one of the most stringent air quality improvement policies in China, one of the countries experiencing the most severe air pollution. China also has a population of approximately 1.4 billion, which is about 18% of the world’s population. This policy evaluation is important because it not only impacts a large segment of the earth’s population, but it might also give some guidance for similar future policies in other developing countries. Our paper’s final contribution is the quantitative measurement of policy interventions based on the policy’s stringency and duration. Identifying effective policy characteristics could be extremely useful to policymakers, but no previous study has attempted to do so.

We discovered that the APPCAP significantly reduced respiratory and circulatory system diseases. The poorest socioeconomic groups and the elderly benefited the most. Although both stringency and duration positively impacted health outcomes, stringency was more effective than duration.

The remainder of this paper proceeds as follows. In Section 2, we explain the policy in detail. In Section 3 and Section 4, we present our data and econometric model. In Section 5, we present the results. In Section 6 and Section 7, we discuss and conclude the paper, respectively.

## 2. Air Pollution Prevention and Control Action Plan (2013–2017)

The State Council of China announced the APPCAP in 2013, and each administrative division implemented its own environmental protection policy. The timing and duration of implementation in the administrative divisions varied greatly. Figure 1 shows the implementation time of the APPCAP at the administrative division level. Beijing and Shanghai, for example, implemented the APPCAP in September and November 2013, respectively, with a four-year policy duration; Sichuan and Guangdong began in February 2014, with a three-year policy duration; and Liaoning and Guangxi began in 2017, with a three-year policy duration. Table 1 shows the enacted time and durations of the APPCAP at the administrative division level.

## 3. Data: China Family Panel Studies

We examined the impact of the APPCAP on health among the population in China. The China Family Panel Studies (CFPS) is a nationally representative study that collects detailed information on people’s health across 25 of 31 administrative divisions in China. From these 25, we excluded Tianjin because it had implemented environmental regulations prior to the APPCAP. Therefore, 24 administrative divisions were used in the analysis (see Table 1). The first CFPS survey was conducted in 2010, with subsequent surveys in 2012, 2014, and 2016 (CFPS2010, CFPS2012, CFPS2014, and CFPS2016). We used information on diagnosis by a doctor over the previous six months to measure the health outcomes. The first wave, which was CFPS2010, unfortunately, did not collect this information. Therefore, we used three waves of the survey, from CFPS2012 to CFPS2016. The earliest APPCAP policy intervention was implemented in 2013 and the latest one in 2017 (see Table 1). Each observation recorded the actual month and year of the survey. The survey took two years to complete. Therefore, CFPS2016 contains answers from 2016 as well as in 2017. Our study period is restricted till 2017 because the environmental protection tax law implemented on 1 January 2018 might bias results by considering a longer time period.

Sample attrition could bias our results. About 17.5% of the 2012 survey respondents did not respond to the 2014 survey, including people who died. The attrition rate was 17% in 2016 from the 2014 survey. We hypothesized that higher mortality in polluted areas could result in higher attrition rates compared to less polluted areas. We ran a regression analysis to check whether the attrition rate was correlated with the pollution level in 2012 using the administrative area as the unit of observation. We did not find any statistically significant result, and the sign was negative, implying that higher polluted areas had a lower attrition rate. We feel confident that sample attrition did not generate bias in our study. 

### 3.1. Dependent Variables

We measured air-pollution-related chronic diseases using CFPS. Based on the literature, we identified two categories of air-pollution-related chronic diseases: respiratory system diseases (asthma, pneumonia, etc.) and circulatory system diseases (hypertension, stroke, etc.) [11,27,28]. 

The CFPS asked the respondents ‘whether (they) had doctor-diagnosed chronic diseases over the past six months’. These questions were asked only for those 16 years old and over. If the answer was yes, the CFPS asked for the details of the diagnosis in the subsequent questions. The CFPS classified diseases, and we constructed a binary variable of respiratory and circulatory disease. The outcome was measured on an average of 23 months after the policy intervention. If the 6 months prior to survey time *t* spanned pre- and post-implementation of the APPCAP, we treated it as a missing value. For example, Sichuan implemented the APPCAP in February 2014, so we treated the survey period from February to September 2014 as missing values. Missing values due to this situation accounted for 6% of the sample.

There is also a concern of self-selection: when survey respondents systematically differ in terms of consulting a doctor and receiving a diagnosis, this can cause bias in the estimates. For example, people in urban areas consult a doctor more frequently than those in rural areas and are more likely to be diagnosed. We checked these possibilities by running a regression: respiratory patients’ rate in 2012 as a dependent variable and pollution level in 2012 as an independent variable in the administrative division level. We did not find any statistically significant result. The result did not change when we used circulatory patients’ rate as a dependent variable. Although we did not find systematic differences across administrative division in terms of doctor-diagnosed chronic conditions, we used only overtime variation with the administrative division fixed effects as a precaution.

### 3.2. Weather Conditions

To isolate the impact of seasonal patterns on health, we additionally controlled for a variety of meteorological factors, including mean temperature, humidity, sunshine, and precipitation during the months in which the survey was conducted [29,30]. Monthly observations of weather conditions were obtained from resources such as the environmental section of China’s Statistical Yearbook from 2013 to 2018 [31,32,33,34,35,36]. The China Statistical Yearbooks report the statistics of the previous year, that is, the 2013 yearbook includes the statistical data from 2012. Weather data are only available for key cities in administrative divisions. Following Shahzad et al. (2020), we used the average measure of cities in administrative divisions [37]. Weather conditions could introduce a measurement error but including weather conditions does not change our results qualitatively.

## 4. The Econometric Model

To estimate the effects of APPCAP on health outcomes, we relied on a difference-in-differences (DID) estimation, which has become a popular way to estimate the causal effect of a specific intervention. DID is a quasi-experimental design using longitudinal data from treatment and control groups. The basic assumption needed is that unobserved differences between the treatment and control groups are the same over time. This assumption can be tested when preintervention data are observed multiple times. Unfortunately, we had only one wave (CFPS2012) before the policy implementation. Therefore, we could not test the parallel trend assumption in our study.

The dependent variable was binary and extremely rare: 1.44% and 6.57% of our 56,958 observations for respiratory and circulatory disease, respectively. As a result, we decided to use logistic regression [38]. 

Our baseline econometric model was as follows:$$P\left(Y_{i}=1|X_{i}\right)=\frac{\exp\left\{\theta \left(X_{i};\beta \right)\right\}}{1+\exp\left\{\theta \left(X_{i};\beta \right)\right\}}$$
where $\theta \left(X_{i};\beta \right)$ can be represented as:$$\theta \left(X_{i};\beta \right)=\beta_{0}+\beta_{1}Treated_{j}\times Post_{t}+\beta_{2}X_{ijt}+\beta_{3}Weather_{jt}+\delta_{j}+\gamma_{t}+\varepsilon_{ijt}$$

Here, the probability that a respondent *i* suffered from a respiratory or circulatory system disease in the previous six months in administrative division j at time *t* by month has been estimated. $Y_{i}$, a binary random variable with mean $E\left(Y_{i}\right)=p$ and variance equal to $p\left(1-p\right)$, indicates if the respondent suffered from doctor-diagnosed chronic diseases over the past six months. $X_{i}$ is a $\left(Κ\times 1\right)$ vector of explanatory variables. $\theta \left(X_{i};\beta \right)$ is referred to as the predictor function, and β is a vector of parameters.

In the regression model, *β*_0_, the constant term, reflects the expected value of the probability of suffering from a respiratory or circulatory system disease when all of the predictor variables in the model are equal to 0. *Treatment_j_* is 1 if the administrative division implemented the APPCAP policy and *Post_t_* is 1 if the time is after the implementation of the APPCAP policy. *β*_1_, the coefficient of *Treatment_j_* ×
*Post_t_,* captures the effect of the APPCAP on health outcomes assuming all other predictor variables are held constant. *Treatment_j_* and *Post_t_* should be included in the model. We included the administrative division fixed effect $\delta_{j}$ and monthly fixed effects $\gamma_{t}$ in our model instead of *Treatment_j_* and *Post_t_*, respectively [39]. Since we used a logistic model coefficient, the marginal effects should be calculated. We report the marginal effects of the policy interventions at the mean as well as the coefficients of the model. 

Age, sex, marital status (never married, married, cohabitating, divorced, and widowed, with never married as the reference group), and education level (primary school, middle or high school, and university graduates or above, with primary as the reference group), employment status, and annual household income were all controlled for as a set of demographic variables, $X_{ijt}$ [29,40]. 

Chen et al. (2013) discovered that the health effect of particulate air pollution varied by season, with the largest effect in winter and summer in China [41]. We controlled for the weather conditions as mentioned in the previous section. $\varepsilon_{ijt}$ is an error term. 

Table 2 presents the descriptive statistics of the variables used in the analyses. Approximately 48% of the sample included were men, and 47% of them resided in a city. The average age of the sample was 49 years.

## 5. Results

First, we analyzed whether the APPCAP influenced the adult sample. As we mentioned in the previous section, chronic disease questions were surveyed only for 16 years old and over. Therefore, we used everyone for whom we had information on chronic disease conditions. The coefficient ($\beta_{1})$ in the logit model did not suggest a marginal effect. The marginal effect at the means of the independent variables was calculated. This is shown in the following tables as marginal effects.

### 5.1. All Adults

The regression results obtained by estimating the equation in Section 4 are presented in Table 3. Columns 1 and 2 show the effect of the APPCAP, adjusted for observable individual characteristics, weather conditions, administrative-division fixed effects, and monthly fixed effects on respiratory and circulatory system diseases after the APPCAP implementation, respectively. The first row *Treatment_j_* ×
*Post_t_* is the coefficient measuring the effect of the policy intervention on the dependent variable in the difference-in-differences model. 

Age was positive and statistically significant for both columns, meaning that as the age of people increased, the probability of being diagnosed with respiratory or circulatory diseases increased. Males showed a higher probability of having respiratory diseases but a lower probability of having circulatory diseases. Drinking or smoking showed negative effects on circulatory diseases. Married people showed a statistically significant increase in the probability of circulatory disease compared to people who were never married. In addition, people who were employed showed a statistically significantly lower probability of circulatory disease compared to those who were unemployed. 

For our main variable of interest, we calculated the marginal effects at means. The APPCAP resulted in a significant decrease in the average probability of being diagnosed with respiratory diseases and circulatory system diseases by 0.499 percentage points and 0.757 percentage points, respectively. In respiratory diseases and circulatory system diseases, the treatment effect of the APPCAP was a 34.6% (=0.499/1.44) and 11.5% (=0.757/6.57) reduction in the sample mean, respectively. According to the findings, the APPCAP resulted in a significant improvement in adult health.

There could be an administrative division-specific time trend. Large cities, for example, may experience faster health care sector growth, and these unobserved factors might bias our results. We ran an additional regression that included administrative division-specific time trends to account for this. Our results remained qualitatively the same even after including an administrative division-specific time trend.

### 5.2. Results by Subsample

#### 5.2.1. Sex and Age

There might be heterogeneity of the policy effects on chronic diseases across different populations. From the sex perspective, there might be differences physically [42]. Specific demographics, particularly older adults, were identified as potentially more susceptible to air pollution effects than the general population because their physiological processes deteriorate with age and their respiratory tract’s ability to resist PM decreases with age [42,43]. We classified respondents into two sex categories and three age groups: young (16 to 39 years), middle-aged (40 to 64 years), and elderly (65 years and over) [44]. As previously stated, a logistic regression approach was employed.

Table 4 shows the effect of the APPCAP on the probability of developing respiratory (upper block) and circulatory system diseases (bottom block) by sex and age. In the first column, we represent the results reported in Table 3. In the second column, males’ risk of respiratory disease was reduced by 0.737 percentage points following APPCAP adoption. This is a statistically significant change and a 46.7% (=0.737/1.578) drop of the sample mean. In the third column, females’ risk of respiratory disease was decreased, but it was not statistically significant, and the magnitude was also 43% compared to that of males.

In the fourth to sixth column, we present the DID coefficients and marginal effects at means by age group. The elderly (65 years old and above) benefited the most. The rate of respiratory disease fell by 1.37 percentage points with APPCAP, and it was statistically significant. Based on the sample mean of 2.5%, there was a 55.5% reduction in the sample mean. 

At the bottom of Table 4, we report the estimation results for circulatory system disease in the same format. The patterns of the results were different. Males’ risk of circulatory disease did not show a statistically significant result. However, the risk for circulatory system diseases reduced by 1.508 percentage points among females, which was 26.6% of the sample mean. In the fourth to sixth column, the age group results also showed a slightly different pattern compared to respiratory diseases. The risk of circulatory system diseases was reduced by 1.11 and 2.95 percentage points or 16.42% and 17.89% of the sample mean for middle aged (40–64) and older adults (>65 years old), respectively. The results by sex can be viewed as consistent with the studies by Granados-Canal et al. (2005), Shin et al. (2020), Su et al. (2016), Xia et al. (2017), Anderson et al. (2003), and Dominici et al. (2006) that males have a higher risk for respiratory diseases, while females have a higher risk for circulatory system diseases from PM_2.5_ [45,46,47,48]. The results by age were also consistent with the literature, since exposure to PM_2.5_ increases the risk for cardiovascular and respiratory diseases among the elderly [49,50].

#### 5.2.2. Socioeconomic Status

In this section, we will examine the heterogeneity across different socioeconomic statuses. People with low income are more likely to work outdoors (for example, in the construction, transportation, and road services industries), and they might suffer from more exposure to air pollution than people with medium or high income [24,42]. In this study, socioeconomic status was defined through educational attainment levels [51]. We divided the respondents into three groups by education level: primary school, middle or high school, and university graduates or higher.

About 26% of the sample had only completed primary school, 30% had completed middle or high school, and 43% had completed a university diploma or above. In the first column, we present the results reported in Table 3. The upper block shows respiratory disease, and the bottom block reports circulatory disease as an outcome. In the second column, we report the results for the low socioeconomic group. There were 0.702 and 1.003 percentage point reductions in respiratory diseases and circulatory system diseases, respectively, for the low socioeconomic group, which made up 44.5% and 12.8% reductions in the sample mean. The reduction in respiratory disease among people with less education was three times higher than the reduction in circulatory disease in the same group.

The medium and high socioeconomic status groups did not show any statistically significant results, as reported in the third and fourth column of Table 5. This might because high socioeconomic status ensures access to more resources that can protect people from increased exposure, such as private transportation versus public, indoor versus outdoor work environments, and better constructed housing [52,53].

### 5.3. Results by Policy Characteristics

Policymakers are interested in which characteristics of a policy determine its effectiveness. In this section, we consider two policy characteristics: the stringency and duration of the policy. The stringency is measured as the difference between the PM_2.5_ level at pre-intervention time and the reduction goal of the PM_2.5_ level. For example, if the pre-intervention level of PM_2.5_ was 65 μg/m^3^, and the goal level was 55 μg/m^3^, then the stringency was recorded as 10 μg/m^3^ (=65 − 55). The duration is measured as the total months of the policy from implementation to the end. If the policy characteristics were correlated with air pollution levels before intervention, then the policy characteristics would be endogenously decided, which would generate bias. Figure 2a shows the relationship between the stringency and air pollution levels of PM_2.5_ before intervention. Figure 2b shows the policy duration and air pollution levels before intervention. We do not see any pattern in these graphs, which means that both policy characteristics were not endogenously decided. 

We ran a regression using stringency or duration as a key independent variable. Table 6 shows the effect of the policy characteristics on the probability of developing respiratory or circulatory system diseases by stringency (columns 1 and 2) and duration (columns 3 and 4). The mean stringency was 14.6. When we ran the regression using stringency, the marginal effect at the mean was −0.027, as shown in column 1 of Table 6. This means that the respiratory disease rate would be lowered by 0.027 percentage points when the policy goal increased by one unit. Considering the average stringency of 14.6, the marginal effects would be −0.394 (0.027 × 14.6), which was similar to what we obtained in Table 3. This result was statistically significant. A stronger policy would improve people’s respiratory conditions. As shown in column 2 of Table 6, the marginal effects for the circulatory disease were −0.07 percentage points with an increase in stringency by one unit. Considering the average stringency of 14.6, the marginal effects would be −1.02 (0.07 × 14.6). This result was statistically significant. A stronger policy would also improve people’s circulatory conditions.

From columns 3 and 4 of Table 6, we see that both respiratory and circulatory system diseases significantly decreased as the duration of the policy intervention increased. The average duration of respiratory disease would decrease by 0.38 (0.011 × 34.9) percentage points (26.6% of the sample mean), and circulatory disease would decrease by 0.59 (0.017 × 34.9) percentage points (9% of the sample mean) after 34.9 months of implementation. The duration of the policy had a higher impact on respiratory diseases.

### 5.4. Placebo Test

We found that the APPCAP statistically significantly reduced the probability of having respiratory or circulatory diseases. We wanted to ensure that our findings were measuring the true effect of policy interventions. The common way of testing this is to perform a placebo test using a fictitious event or treatment as a falsification strategy [54]. For example, if we provide policy intervention timing as fake information or conduct a falsification exercise using non-air-pollution-related diseases, the model should not show a statistically significant result [55,56]. 

#### 5.4.1. In-Time Placebo Test

The first placebo test was to assume the policy intervention happened 6 months before the actual policy intervention; the results are presented in the upper block of Table 7. The second test was to assume that the policy intervention happened one year before the true intervention; the results are in the middle block of Table 7. For example, Shannxi implemented the policy in December 2013; however, we coded the implementation times for Shannxi as June 2013 and December 2012 for the first and second tests, respectively. None of the coefficients were statistically significant, and the magnitudes of the four estimates were very small compared to Table 3. 

#### 5.4.2. Placebo Outcome Test

In addition to the placebo test using the fake policy intervention time, one can also conduct a falsification exercise using the health outcomes that unaffected by ambient air pollution [57]. We considered non-air-pollution-related diseases as outcome variable. Non-air-pollution-related diseases takes the value of 1 if the respondent was diagnosed with any type of disease besides respiratory and circulatory system diseases in the recent six months, and 0 otherwise. The results are shown in the bottom block of Table 7. The results are statistically insignificant, which indicates that APPCAP had no significant effect on non-air-pollution-related diseases.

## 6. Discussion

This study evaluated the impacts of environmental regulation on chronic health conditions using the APPCAP, the first nationwide and very stringent environmental regulation in China. Our identification strategy was a difference-in-differences model using different implementation timing of the APPCAP by administrative divisions. We used the China Family Panel Studies, a representative survey of China, to measure chronic conditions using doctor-diagnosed diseases over the last six months. We discovered that the APPCAP reduced respiratory and circulatory system diseases by 34.6% and 11.52%, respectively, in all adults. Our study could serve as a reference for developing countries that are experiencing serious air pollution problems when formulating environmental preservation policies.

We cannot compare our estimates with the literature directly due to the very limited literature measuring chronic conditions. However, we can compare the magnitude with mortality, which is the most studied outcome. Yue et al. (2020) found that the APPCAP reduced mortality in 2017 by 6.8% compared to 2013 [58]. Compared to our estimates of a 34.6% reduction in respiratory and an 11.5% reduction in circulatory system diseases, the mortality reduction was small in magnitude. However, considering that few people die among those who suffer from chronic conditions, this magnitude of difference appears reasonable. Liang et al. (2019) estimated that acute exacerbations of chronic obstructive pulmonary disease (COPD) changed from 12,679 in 2013 to 7377 cases in 2017 in China [10]. This number was calculated from the increase in PM_2·5_ pollution above the expected number of cases if daily PM_2·5_ concentrations had not exceeded the WHO target (25 μg/m^3^). Their results were a 41.8% ((7377–12,679)/12,679) reduction, and our estimates are in the same ballpark.

Our study had several limitations. First, the outcome was measured based on self-reporting rather than objective records, such as medical records or reimbursement records from a health insurance company. However, the outcome, doctor-diagnosed disease during the last six months, had some objective features unlike other types of self-reported answers, although it can still be affected by possible recall bias. Another issue related to data was that we used a sample not the population. The sample had a sampling error, and the standard errors related to the coefficients considered this. 

Second, we measured the total effects without isolating the spatial spillover effects. Fang et al. (2019) discovered negative spillover effects in their study when Beijing, Tianjin, and Hebei implemented a clean air act [18]. The pollution level in the target area decreased at the cost of increased pollution levels in nearby areas, suggesting that pollution emission sources moved from the target area to nearby areas. We evaluated only the total average effects of the policy intervention. However, the policy intervention we evaluated has been implemented throughout most of China. Negative spillover might have occurred in the short run, but nearby areas were also impacted by the policy eventually.

Third, we could not measure outcomes for children. Many previous studies found an increased prevalence of asthma among children who were more susceptible physically [59]. The CFPS unfortunately did not ask about chronic conditions for children. 

Fourth, there might have been other changes related to air pollution control during the period that we considered. For example, the State Council of China implemented the environmental protection tax law on 1 January 2018. The timing of the implementation differed from the APPCAP, and the environmental protection tax law replaced the pollutant discharge fee that had been in effect for the past 40 years [60]. China planned to increase the proportion of non-fossil fuels in primary energy consumption by about 20% by 2030 compared with 2005 to comply with the United Nations Framework Convention on Climate Change released in 2015 [61]. However, no relevant law was enacted during our study period 2013–2017 with the same time schedule as APPCAP. To the best of our knowledge, there was no other environmental policy confounding with APPCAP that biased our estimates. 

Fifth, we measured the health benefits of air quality improvement. Although low educated people or aged population received biggest health benefits, we cannot exclude other people enjoying it in the same region due to public good nature of air quality. Therefore, we face limited policy recommendation based on benefit assessment that we do here. More targeted policies can be designed in terms of cost assessment such as how to reduce air pollutants less costly. However, our research still makes valuable contributions for policy makers. For example, the aging population that China is facing makes the benefits of air quality improvement greater in the future.

## 7. Conclusions

Assessing the impact of past air quality regulatory policies on public health provides a solid basis for the decision-making process used to review the effectiveness of past regulations and aids in causality inference and the development of future policies [62].

We examined the effects of the nationwide air pollution control policy in China on chronic health conditions among adults using the representative survey, CFPS. We also used a concrete econometric technique to find the causal effect with a quasi-experimental situation. We found that the air pollution control policy statistically significantly reduced respiratory and circulatory chronic diseases among adults, which has not been studied previously. Lowering air pollution cannot be achieved for free. Crane and Mao (2015) estimated that replacing coal with natural gas for residential and commercial heating could cost USD 32 billion to 52 billion and replacing half of China’s coal-fired electric power generation with renewables or nuclear power could cost USD 184 billion [63]. When we considered the benefits of the air pollution control policy, our results found highly positive benefits, which were not considered previously. Moreover, our study suggests that stringency was more effective than duration. Although the APPCAP (2013–2017) was considered the most stringent air pollution control policy in China, PM_2.5_ concentration in 2018 was still almost five times higher than the WHO target value of 10 μg/m^3^. China needs to continue air pollution reduction policies to achieve the WHO standard of 10 μg/m^3^ for PM_2.5_ and these policies will improve the chronic condition of people. 

## Figures and Tables

**Figure 1 ijerph-19-12584-f001:**
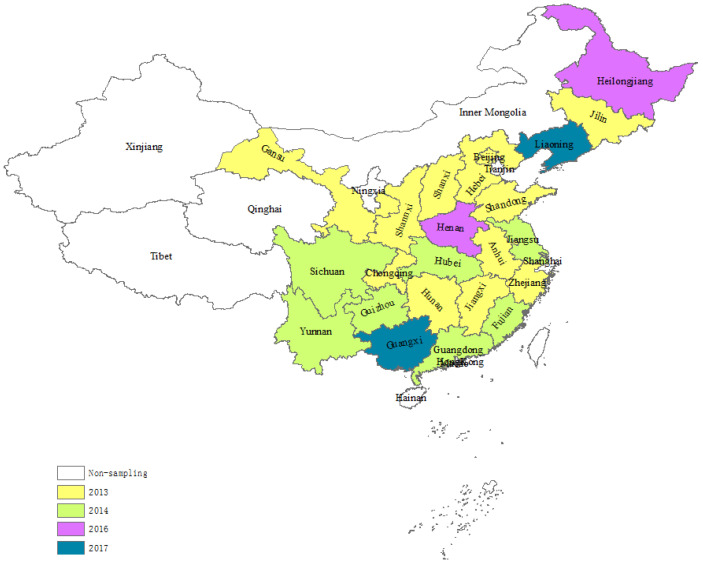
APPCAP implementation time by year. *Source*: Official website of each administrative division.

**Figure 2 ijerph-19-12584-f002:**
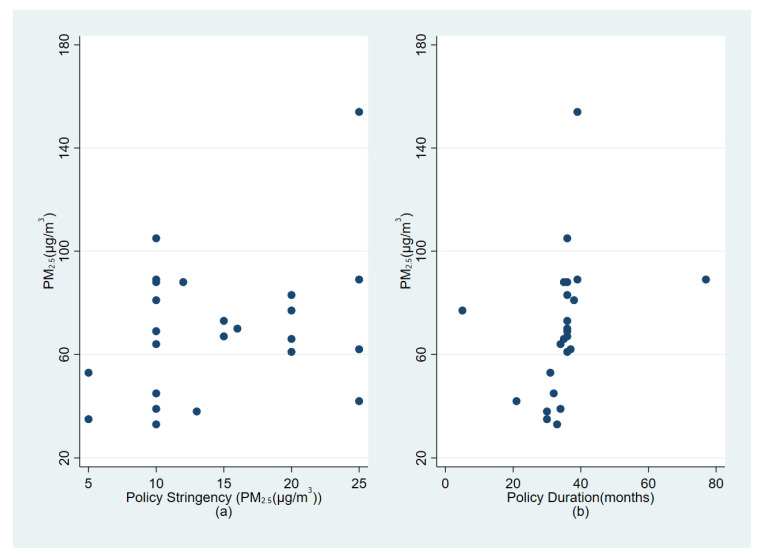
Policy characteristics and air pollution levels before the intervention. *Notes*: The *y*-axis represents PM_2.5_ concentrations in the year before policy intervention. (**a**) The *x*-axis represents the policy stringency measured by the difference between the pre-intervention level of PM_2.5_ and the goal level of PM_2.5_. For example, the pre-intervention level of PM_2.5_ for Sichuan was 65 μg/m^3^, and the goal level of PM_2.5_ was 55 μg/m^3^, which is 10 μg/m^3^ (=65 − 55); (**b**) the *x*-axis represents the duration of the policy in months.

**Table 1 ijerph-19-12584-t001:** APPCAP implementation time by month and duration.

Administrative Division	APPCAP Enacted Time	Duration
Shandong	July 2013	~2020
Beijing	September 2013	~2017
Hebei	September 2013	~2017
Shanxi	October 2013	~2017
Shanghai	November 2013	~2017
Anhui	December 2013	~2017
Chongqing	December 2013	~2017
Shannxi	December 2013	~2017
Jilin	December 2013	~2017
Zhejiang	December 2013	~2017
Jiangxi	December 2013	~2017
Hunan	December 2013	~2017
Gansu	December 2013	~2017
Hubei	January 2014	~2017
Jiangsu	January 2014	~2017
Guangdong	February 2014	~2017
Sichuan	February 2014	~2017
Yunnan	March 2014	~2017
Guizhou	May 2014	~2017
Fujian	June 2014	~2017
Heilongjiang	March 2016	~2018
Henan	July 2016	~2017
Liaoning	April 2017	~2020
Guangxi	June 2017	~2020

*Source*: Official website of each administrative division. *Notes*: Thirty-one administrative divisions implemented their own environmental protection policies. The survey data at the individual level covered 25 administrative divisions. From these 25, we excluded Tianjin because it had implemented environmental regulations prior to the APPCAP. Therefore, 24 administrative divisions were used in the analysis.

**Table 2 ijerph-19-12584-t002:** Descriptive statistics: CFPS2012, 2014, 2016.

Variables	Mean	Standard Deviation
**Dependent Variables**		
Respiratory diseases (%)	1.44	0.12
Circulatory system diseases (%)	6.57	0.17
**Independent Variables**		
*Weather Conditions*		
Mean Temperature (°C)	21.62	9.48
Humidity (%)	71.99	8.69
Precipitation (mm)	127.57	109.72
Sunshine (hour)	181.89	53.06
*Demographic characteristics of CFPS surveys*		
Age (year)	48.66	14.81
Mean annual household income ^a^ (log form)	7.14	4.60
Male (%)	48.09	0.50
Urban (%)	46.63	0.50
Labor force participation (%)	72.80	0.44
Primary school or less (%)	26.67	0.46
Middle/high school (%)	29.93	0.46
University or above (%)	43.40	0.49
Married (%)	85.69	0.35
Smoke (%)	29.84	0.46
Drink (%)	16.29	0.37
Coal (%)	6.67	0.25
Observations	56,958	

*Note*: ^a^ Log transformed annual household income (CNY).

**Table 3 ijerph-19-12584-t003:** Logit estimates: effects of the APPCAP on chronic conditions.

	Air-Pollution-Related Diseases	
Variables	Respiratory	Circulatory
**Treatment × Post**	**−0.489 ****	**−0.239 ****
	**(0.192)**	**(0.099)**
Age	0.027 ***	0.085 ***
	(0.004)	(0.003)
Male	0.611 ***	−0.179 ***
	(0.111)	(0.065)
Urban	−0.002	−0.003
	(0.100)	(0.057)
Alcohol	−0.319	−0.316 ***
	(0.132)	(0.077)
Tobacco	−0.681	−0.329 ***
	(0.120)	(0.069)
Coal heating	−0.017	−0.020
	(0.187)	(0.092)
Married	−0.151	0.868 ***
	(0.225)	(0.245)
Cohabitating	0.967	0.994 **
	(0.621)	(0.483)
Divorced	0.227	1.122 ***
	(0.403)	(0.322)
Widowed	−0.079	0.583 **
	(0.297)	(0.263)
Employed	−0.091	−0.318 ***
	(0.109)	(0.058)
ln (Annual household income)	0.006	0.007
	(0.010)	(0.005)
Middle/high school	−0.133	0.014
	(0.126)	(0.068)
University or above	−0.053	0.009
	(0.207)	(0.122)
Constant	−6.623 ***	−8.995 ***
	(1.129)	(0.683)
Observations	56,958	56,958
**Marginal effects at means for Treatment × Post**	**−0.499 ****	**−0.757 ****
	**(0.196)**	**(0.312)**
Dependent variable mean (×100)	1.44	6.57

Notes: The sample included all respondents in CFPS 2012, 2014, and 2016 in Table 2. The dependent variable was respiratory diseases and circulatory system diseases; if the respondent had suffered from respiratory diseases or circulatory system diseases in the previous six months, then the variable was coded as a dummy variable, *respiratory diseases* or *circulatory system diseases* (=1). Treatment × Post is our variable of interest representing difference-in-difference estimates. Weather was controlled for. Administrative divisions and monthly fixed effects are included in the regressions. Clustered standard errors across survey respondents are presented in parentheses. **, and *** indicate significance at the 10%, 5%, and 1% levels, respectively.

**Table 4 ijerph-19-12584-t004:** Logit estimates: effects of APPCAP on chronic conditions, by sex and age group.

**Respiratory System Disease**						
		**Sex**		**Age**		
	**Total**	**Male**	**Female**	**16–39**	**40–64**	**65–**
Treatment × Post	−0.489 ** (0.192)	−0.725 ** (0.296)	−0.338 (0.259)	−0.436 (0.399)	−0.421 (0.269)	−0.766 * (0.408)
Observations	56,958	27,855	29,015	15,906	32,179	8649
**Marginal effects at means for Treatment × Post**	−0.499 ** (0.196)	−0.737 ** (0.30)	−0.319 (0.244)	−0.259 (0.235)	−0.410 (0.260)	−1.367 * (0.718)
Dependent variable mean (×100)	1.443	1.578	1.315	0.965	1.400	2.464
Circulatory System Disease						
		**Sex**		**Age**		
	**Total**	**Male**	**Female**	**16–39**	**40–64**	**65–**
**Treatment × Post**	**−0.239 **** **(0.099)**	**0.043** **(0.150)**	**−0.434 ***** **(0.132)**	**0.303** **(0.480)**	**−0.275 **** **(0.119)**	**−0.286 *** **(0.165)**
Observations	56,958	27,855	29,103	15,906	32,179	8649
**Marginal effects at means for Treatment × Post**	−0.757 ** (0.312)	0.120 (0.419)	−1.508 *** (0.459)	−0.121 (0.188)	−1.109 ** (0.519)	−2.948 ** (1.700)
Dependent variable mean (×100)	6.570	7.432	5.669	0.689	6.760	16.490

*Notes*: The sample included all respondents in CFPS 2012, 2014, and 2016 in Table 2. Sex, age, urban, marital status (married, cohabitating, divorced, and widowed), employed, logarithm form of annual family income, and weather information were included as independent variables. Treatment × Post is our variable of interest representing difference-in-difference estimates. Administrative divisions and monthly fixed effects are included in the regressions. Clustered standard errors across survey respondents are presented in parentheses. *, **, and *** indicate significance at the 10%, 5%, and 1% levels, respectively.

**Table 5 ijerph-19-12584-t005:** Logit estimates: effects of APPCAP on chronic conditions, by education level.

**Respiratory System Disease**				
		**Educational Attainment Levels**		
	**Total**	**Primary**	**Middle**	**College and above**
**Treatment × Post**	**−0.489 **** **(0.192)**	**−0.702 *(0.379)**	**−0.487** **(0.398)**	**0.232** **(0.473)**
Observations	56,958	14,841	16,867	24,361
**Marginal effects at means for Treatment × Post**	**−0.499 **** **(0.196)**	**−0.702 *** **(0.377)**	**−0.343** **(0.278)**	**0.269** **(0.547)**
Dependent variable mean (×100)	1.443	1.579	1.119	1.584
**Circulatory System Disease**				
		**Educational Attainment Levels**		
	**Total**	**Primary**	**Middle**	**College and above**
**Treatment × Post**	**−0.239 **** **(0.099)**	**−0.246 *** **(0.173)**	**−0.045** **(0.204)**	**0.229** **(0.317)**
Observations	56,958	15,188	16,986	24,578
**Marginal effects at means for Treatment × Post**	**−0.757 **** **(0.312)**	**−1.003 *** **(0.705)**	**−0.099** **(0.444)**	**0.753** **(1.046)**
Dependent variable mean (×100)	6.570	7.787	4.559	7.210

*Note*: See notes in Table 4. *, and ** indicate significance at the 10%, and 5% levels, respectively.

**Table 6 ijerph-19-12584-t006:** Logit estimates: stringency of APPCAP on chronic conditions.

	Air-Pollution-Related Diseases			
	Respiratory	Circulatory	Respiratory	Circulatory
**Stringency**	**−0.026 ***** **(0.009)**	**−0.022 ***** **(0.005)**		
**Duration**			**−0.011 **** **(0.004)**	**−0.005 **** **(0.002)**
Observations	56,958	56,958	56,958	56,958
**Marginal effects at means**	**−0.027 ***** **(0.010)**	**−0.070 ***** **(0.016)**	**−0.011 **** **(0.005)**	**−0.017 **** **(0.008)**
Dependent variable mean (×100)	1.443	6.570	1.443	6.570

*Note*: See notes in Table 4. **, and *** indicate significance at the 5%, and 1% levels, respectively.

**Table 7 ijerph-19-12584-t007:** Placebo Test.

	Air-Pollution-Related Diseases	
	Respiratory	Circulatory
** Treatment × Post (6-month in advance)**	**0.152** **(0.426)**	**−0.003** **(0.202)**
Observations	30,111	30,111
**Marginal effects at means for Treatment × Post**	**0.00130** **(0.00365)**	**−0.00007** **(0.00532)**
** Treatment × Post (1-year in advance)**	**0.138** **(0.333)**	**−0.160** **(0.140)**
Observations	26,899	26,899
**Marginal effects at means for Treatment × Post**	**0.00121** **(0.00291)**	**−0.00442** **(0.00387)**
	**Non-air-pollution-related diseases**	
** Treatment × Post**	**0.061** **(0.073)**	
Observations	56,958	
**Marginal effects at means for Treatment × Post**	**0.00409** **(0.00487)**	

*Notes*: See notes in Table 4.

## Data Availability

The raw data used in this article are available upon request to corresponding author.

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
