# Peer review of "Environmental Regulation and Chronic Conditions: Evidence from China’s Air Pollution Prevention and Control Action Plan"

_ijerph, 2022, doi:10.3390/ijerph191912584_

Round 1
Reviewer 1 Report
The manuscript “Environmental Regulation and Chronic Condition: Evidence from China’s Air Pollution Prevention and Control Action Plan” is a very interesting article and provides a valuable information about the impact of China's Air Pollution Prevention and Control Action Plan policy on health outcomes under different socioeconomic and weather conditions. Although similar studies have been carried out on the Chinese mainland yet the present study looks interesting due to unique outcomes. The methods appeared appropriate, but the results and discussion section need to be strengthened more. The abstract and conclusion section of manuscript should be revised more in terms of highlighting the key findings of the study. There are many sentences also that are too long resulting to a bit of confusion (for example sentence “Low-income are ….. air pollution”. Author must revise and rephrase these sentence in more influential way. Further above-suggested work is required to improve the manuscript along with the following comments should be taken into consideration by the authors before the acceptance for publication.
· What is OCED and radiation area? Authors must describe it.
· Figure 2 is not clear. Replot it.
· Sentences “Second due …. Air pollution wordwide” and “The highest ……. from 2005” need proper references. Kindly provide appropriate reference to justify the sentence.
· Sentences “The remainder…. Respectively”. Revise the whole paragraph.
· Figure 3, what is unit of X axis should be mention. Revise legends of Figure 3 accordingly.
· In equation, what is the significance of coefficient β0 β1 β2. Authors must be describe it.
· Since in this paper, a large survey data sets are used which itself have a significant uncertainty. So, uncertainty analysis for used model must be included and discussed in the manuscript.
· Author must discuss about the “treatment effects and difference-in-difference model” briefly.
· Discussion about the table 8, 9, 10 is not satisfactory. Author should revise it and compare with previous similar studies.
Reviewer 2 Report
ijerph-1811062
Title: Environmental Regulation and Chronic Condition: Evidence from China’s Air Pollution Prevention and Control Action Plan
Type: Article
Dear Authors,
The manuscript used the China Family Panel Studies and a difference-in-differences model to investigate the impact of APPCAP on health. The findings showed that the APPCAP significantly reduced illness of the respiratory and circulatory systems.
The introduction section lacks a summary of the established studies.
The manuscript study area is not clear. According to 2. Air Pollution and Control Action Plan (2013~2017), the study area of this manuscript is 24 provincial units. However, the data in 3.2. Weather Conditions are only available for major cities from the China Statistical Yearbook.
Page 9~10 Section 5.2 is an analysis of subsamples, why is the corresponding section 5.1 an adult sample instead of a full sample?
The section 5.Results is too simple for the interpretation of the estimated results, which is basically a restatement of the data, without clarifying the reasons.
The manuscript was not tested for parallel trends, which leaves the applicability of the DID method unprotected. No placebo test was performed in the robustness test section.
How can results in 5.4.1 and 5.4.2 that are not statistically significant be considered robust? The robustness of the estimation results should be confirmed at least at the sign and significance level of the coefficients. In addition, the idea of testing whether APCAP has an effect on diseases other than respiratory and circulatory system diseases is problematic.
The conclusion section needs improvement. There are no targeted policy recommendations based on the basic estimation results of the article, especially the heterogeneity estimation results, which leads to the lack of relevance.
The manuscript has some basic formatting problems. For example, the references in Page 3 are too large in font size.
Reviewer 3 Report
Dear Authors,
Thank you for your very interesting paper!
Before publishing it needs revisions:
- Abstract: No too short. Please add when study was carried out? With what kind of data you used?
- References in the text: e.g. page 1; no scientific references at all?
- Also reference style; please check it. I guess that currently it is not suitable for the Journal.
- Moreover, please check is all references used in the text also in the Reference list, e.g. page 3, China Statistical Yearbook 2014-2018.
- Lay-out of Tables: e.g. Table 3 and 4. Their lay-out and titles must be improved.
- 5.2.1. Used categories: young 16-45 years old? Is citizen of more than 40 years old young?
- Reference list: I think that all related references have not been used because your Reference list is so short. - Less than 20 references.
- The paper needs more scientific sound.
Reviewer 4 Report
This paper used the data of China Family Panel Studies (CFPS) to investigate the impact of Air Pollution Prevention and Control Action Plan (APPCAP) on health. The subject of the research is interesting and the conclusion has policy implications. However, there are still some imperfections need to be revised as following.
(1) In the introduction part, since this paper studies the issue of China, it is redundant to describe the current situation of air pollution in India.
(2) The impact mechanism of environmental regulation on human health needs to be discussed in depth. It is too intuitive to simply believe that the environmental regulation improves environmental quality and is therefore conducive to human health, and this empirical study does not take environmental quality as an intermediary variable.
(3) The abstract said that this paper used a difference-in-differences model. But in the econometric model, APPCAP is just a dummy variable and we cannot see the DID method from the model specification.
(4) Environmental regulation is a concept with a wide extension, especially during the sample period China has introduced many environmental related policies (such as the introduction of environmental protection tax). Whether it is appropriate to reflect environmental regulation only through a single policy (APPCAP) needs further consideration and discussion.
(5) Whether the research conclusion has policy implications for other developing countries can be further discussed.
Reviewer 5 Report
The paper “Environmental Regulation and Chronic Condition: Evidence from China’s Air Pollution Prevention and Control Action Plan” focuses on a current and very important issue. The findings obtained could be significant, but the research should be carried out over a longer period of time. The short time period is not a representative sample, especially with regard to air pollution and its effects on the human body or health. There is always a time shift in such a correlations. It is therefore debatable whether the period 2013-2017 is a sufficient sample.
The following changes are required:
- - Page 2, “Air Pollution Prevention and Control Action Plan” (APPCAP) – there is no need to translate the abbreviation once again,
- - Figure 2 –the y axis descriptions need to be corrected,
- - Note the size of the font and spaces throughout the manuscript,
- - The personal form should be avoided in scientific studies, passive time is recommended,
- - the first acapit - the description of figure 2 should be as close as possible to the figure,
- - it is recommended to present the percent as a %
- - figure 4 - put units on the x-axis, it should be explain what the scale on the x-axis means in the case of c "(c) The x-axis represents the strength of the strategy",
- - page 5, second paragraph - “We confirmed that the policy characteristics correlated with the preintervention pollution level measured by PM”. It should be explain on what basis the above conclusion was formulated, as it was rightly noticed, it does not follow from Figure 4,
- - It is recommended to expand the bibliographic part.
Round 2
Reviewer 2 Report
Although the revised manuscript has been improved, there are still some key problems to be solved.
(1) For comment " How can results in 5.4.1 and 5.4.2 that are not statistically significant be considered robust? The robustness of the estimation results should be confirmed at least at the sign and significance level of the coefficients. In addition, the idea of testing whether APCAP has an effect on diseases other than respiratory and circulatory system diseases is problematic.", the authors did not solve but deleted.
That is to say, the authors admitted that the previous Table 9 cannot prove the robustness of the empirical conclusions, which is a fatal flaw in this research.
(2) The authors mentioned in response that "Although we cannot provide clear policy recommendations, ...", which can lead to the lack of realistic significance. If an empirical study is carried out on practical problems, but the policy recommendations cannot be summarized according to the conclusions, then the research has no significance and value.
(3) In addition, there are still some formatting issues. For example, × and * are mixed in Table 5.
Please give a detailed explanation of the above problems.
Author Response
We attached responses in the file

Reviewer 4 Report
This paper can be accepted in present form.
Author Response
No comments to reply
Reviewer 5 Report
The authors did not take into account the very important comment of the reviewer. It is highly debatable whether reliable research results can be obtained based on 5-year data. Of course, statistical tools make it possible to represent some dependencies in the data. But, I emphasize once again, the short period of time is not a representative sample, especially with regard to air pollution and its effects on the human body. The time shift is also an additional factor. Therefore, the question arises as to how reliable the obtained test results are?
Author Response
We are very sorry that we did not take into account your very important comment related to short time span. My sincere apologies for missing your point in the first revision.
We attached reponses to your points in the file

Round 3
Reviewer 2 Report
The policy recommendations are still written only on the surface and need to be more specific, which can also be proposed targeted based on the empirical results.
The rest of the manuscript has been revised well.
Author Response
We attached the responses.
